# An efficient CRISPR-based strategy to insert small and large fragments of DNA using short homology arms

Oguz Kanca[1,2], Jonathan Zirin[3,4], Jorge Garcia-Marques[5], Shannon Marie Knight[3,4], Donghui Yang-Zhou[3,4], Gabriel Amador[3,4], Hyunglok Chung[1,2], Zhongyuan Zuo[1,2], Liwen Ma[1], Yuchun He[1,6], Wen-Wen Lin[1], Ying Fang[1], Ming Ge[1], Shinya Yamamoto[1,2,7,8], Karen L Schulze[1,2,6], Yanhui Hu[3,4], Allan C Spradling[9], Stephanie E Mohr[3,4], Norbert Perrimon[3,4], Hugo J Bellen[1,2,6,7,8]*

[1]Department of Molecular and Human Genetics, Baylor College of Medicine, Houston, United States; [2]Jan and Dan Duncan Neurological Research Institute, Texas Children's Hospital, Houston, United States; [3]Howard Hughes Medical Institute, Harvard Medical School, Boston, United States; [4]Department of Genetics, Harvard Medical School, Boston, United States; [5]Janelia Research Campus, Howard Hughes Medical Institute, Ashburn, United States; [6]Howard Hughes Medical Institute, Baylor College of Medicine, Houston, United States; [7]Program in Developmental Biology, Baylor College of Medicine, Houston, United States; [8]Department of Neuroscience, Baylor College of Medicine, Houston, United States; [9]Department of Embryology, Howard Hughes Medical Institute, Carnegie Institution for Science, Baltimore, United States

*For correspondence: hbellen@bcm.edu

**Abstract** We previously reported a CRISPR-mediated knock-in strategy into introns of *Drosophila* genes, generating an *attP-FRT-SA-T2A-GAL4-polyA-3XP3-EGFP-FRT-attP* transgenic library for multiple uses (Lee et al., 2018a). The method relied on double stranded DNA (dsDNA) homology donors with ~1 kb homology arms. Here, we describe three new simpler ways to edit genes in flies. We create single stranded DNA (ssDNA) donors using PCR and add 100 nt of homology on each side of an integration cassette, followed by enzymatic removal of one strand. Using this method, we generated GFP-tagged proteins that mark organelles in S2 cells. We then describe two dsDNA methods using cheap synthesized donors flanked by 100 nt homology arms and gRNA target sites cloned into a plasmid. Upon injection, donor DNA (1 to 5 kb) is released from the plasmid by Cas9. The cassette integrates efficiently and precisely in vivo. The approach is fast, cheap, and scalable.
DOI: https://doi.org/10.7554/eLife.51539.001

## Introduction

A main goal of the Drosophila Gene Disruption Project (GDP) is to create genetic tools that facilitate an integrated approach to analyze the function of each gene in detail. This involves assessment of the loss of function phenotype, identification of the cells that express the gene, determination of the subcellular protein localization, selective removal of the transcript or protein in any tissue, the ability to perform immunoprecipitation of the protein and its interacting proteins or DNA, rescue of the induced fly mutant phenotypes with fly or human cDNAs and assessment of the consequences of amino acid variants in vivo.

These elegant and precise manipulations are made possible by the integration of a Swappable Integration Cassette (SIC) in the gene of interest (GOI) using transposon mediated integration (*Minos*-mediated Integration Cassette [MiMIC]; *Venken et al., 2011*; *Nagarkar-Jaiswal et al., 2015a*) or homologous recombination mediated by CRISPR, a technique we named CRIMIC [CRISPR-mediated Integration Cassette] [*Zhang et al., 2014*; *Diao et al., 2015*; *Lee et al., 2018a*]). A SIC is typically flanked with *attP* sites and can be replaced using Recombinase Mediated Cassette Exchange (RMCE) (*Bateman et al., 2006*; *Venken et al., 2011*). The CRIMIC variety of SIC currently used by the GDP is an artificial exon consisting of *attP-FRT-SA-T2A-GAL4-polyA-3XP3-EGFP-FRT-attP* inserted in a coding intron (intron flanked by two coding exons) of the GOI (*Lee et al., 2018a*). This insert typically creates a severe loss of function allele and generates a GAL4 protein that is expressed in the target gene's spatial and temporal expression pattern (*Diao et al., 2015*; *Gnerer et al., 2015*; *Lee et al., 2018a*). The resulting GAL4 can then be used to drive a *UAS-nuclear localization signal (NLS)::mCherry* to determine which cells express the gene or a *UAS-membrane (CD8)::mCherry* to outline the cell projections (*Brand and Perrimon, 1993*; *Shaner et al., 2004*). Alternatively, a *UAS-GOI cDNA* can be used to test for rescue of the loss of function phenotype induced by the insertion cassette. This provides a means for rigorous quality assessment of the genetic reagent and, when combined with mutant and/or truncated forms of the *UAS-GOI cDNA,* facilitates structure-function analysis. In addition, a *UAS-human-homologue cDNA* of the GOI permits humanization of the flies and assessment of human variants (*Bellen and Yamamoto, 2015*; *Kanca et al., 2017*; *Şentürk and Bellen, 2018*; *Chao et al., 2017*; *Yoon et al., 2017*).

The SIC can also be replaced by an artificial exon that consists of *SA-Linker-EGFP-FlAsH-StrepII-TEV-3xFlag-Linker-SD*, abbreviated *SA-GFP-SD,* which adds an integral Green Fluorescent Protein (GFP) and other tags to the gene product (*Venken et al., 2011*; *Nagarkar-Jaiswal et al., 2015a*). This tag does not disrupt protein function in 75% of cases examined and permits the determination of the subcellular protein localization (*Venken et al., 2011*; *Nagarkar-Jaiswal et al., 2015a*; *Yoon et al., 2017*; *Lee et al., 2018a*) as well as removal of the protein in any tissue using specific GAL4 drivers (*Jenett et al., 2012*) to drive a DeGradFP protein that leads to polyubiquitination and degradation of the protein of interest (*Caussinus et al., 2012*; *Nagarkar-Jaiswal et al., 2015a*; *Lee et al., 2018b*). The GFP tag can also be used as an epitope for immunoprecipitation to determine interaction partners of the tagged protein (*Neumüller et al., 2012*; *Zhang et al., 2013*; *David-Morrison et al., 2016*; *Yoon et al., 2017*).

Additionally, SICs can be replaced by other RMCE vectors that enable integration of additional binary or tertiary systems (*e.g. SA-T2A-LexA, SA-T2A-split GAL4*; *Gnerer et al., 2015*; *Diao et al., 2015*) to obtain finer tools to express transgenes in specific cell populations. SICs can also be used to generate conditional alleles of targeted genes (Flip-flop and FLPstop; *Fisher et al., 2017*; *Nagarkar-Jaiswal et al., 2017*). Finally, strategies have been developed to convert SICs through genetic crosses rather than by injection (*Trojan Exons* and *Double Header*; *Nagarkar-Jaiswal et al., 2015b*; *Diao et al., 2015*; *Li-Kroeger et al., 2018*).

Although the above reagents form a powerful toolset, the generation of libraries of many thousands of genes based on these methods is labor-intensive and costly. The cost for reagents and labor for the generation of a single CRIMIC fly line is $1,000–2,000. Indeed, to create each CRIMIC construct we need to amplify two 1 kb homology arms, clone these arms on either site of a SIC in a plasmid, sequence verify the constructs, amplify and inject the DNA with a target-specific gRNA into 600 embryos, screen to obtain several independent transgenic flies, establish several fly stocks for each construct, PCR-verify the insertions, and cross each line with *UAS-mCherry* to determine expression patterns. Since the production of each transgenic line involves multiple steps, low failure rates at each step accumulate and decrease the overall success rate to ~50%. Given that we are in the process of tagging ~5000 genes that contain suitable introns, it is highly desirable to develop a more efficient, less labor-intensive, and cheaper alternative. One of the main bottlenecks is the production of large (5 kb) SIC homology donor plasmids containing a visible dominant marker and flanked by two ~ 1 kb homology arms to promote homologous recombination (*Beumer et al., 2008*; *Beumer et al., 2013*; *Bier et al., 2018*; *Lee et al., 2018a*). We therefore explored a series of alternative strategies to reduce the construct size and facilitate cloning.

Here, we report the development of methods, using either a PCR-generated, single stranded DNA donor (ssDNA drop-in) or a synthesized double stranded homology donor (dsDNA drop-in) that greatly simplify the generation of homology donor constructs and improve the transgenesis rate. We tested both methods in vivo in *Drosophila* and targeted the same 10 genes with five different constructs to assess transformation efficiency and accuracy of integration. We show that the ssDNA drop-in method works efficiently in *Drosophila* S2R+ cells for constructs that are less than 2 kb and we used this method to mark several cellular organelles with GFP tagged proteins. The dsDNA drop-in strategy is based on short homology arms flanking SICs of up to 5 kb. The success rate for tagging the tested genes was 70–80%. The dsDNA drop-in donor vector is easy and cheap to produce, transformation efficiency is high, and the insertions are precise. Hence, these changes significantly decrease the costs of generating a transgenic CRIMIC library and make the CRIMIC technique more accessible to others. We anticipate that the drop-in approaches will also be useful in other species.

## Results and discussion

### ssDNA homology donors

dsDNA homology donors for insertion of large cassettes in *Drosophila* typically require stretches of 500 nt to 1 kb of homology to the target site on either side of the SIC (*Rong and Golic, 2000*; *Beumer et al., 2013*; *Zhang et al., 2014*; *Diao et al., 2015*; *Bier et al., 2018*; *Lee et al., 2018a*). The large size of the homology regions affects cloning efficiency of the donor constructs (*Lee et al., 2018a*). Single stranded homology donors typically rely on much shorter homology arms (50–100 nt) to successfully integrate short DNA segments in *Drosophila* (~200 nt) based on homology-directed repair (*Beumer et al., 2013*; *Gratz et al., 2013*; *Wissel et al., 2016*; *Bier et al., 2018*). To facilitate and speed up the preparation of homology donor constructs, we decided to test ssDNA donors for CRISPR-mediated homologous recombination of donors that are 1 to 2 kb.

To produce ssDNA homology donors, we established a cloning-free method based on PCR (*Higuchi and Ochman, 1989*). We generated SIC containing PCR templates that can be amplified with an M13 universal primer-derived primer pair (26 nt forward primer and 24 nt reverse primer). Gene-specific 100 nt homology arms are incorporated into these primers as 5' overhangs (left homology arm to the forward primer and reverse complement of right homology arm to the reverse primer; *Figure 1A*). One of the primers is phosphorylated at its 5' end. The resulting PCR product contains a 5' phosphorylated strand and a non-phosphorylated strand. When this PCR product is treated with Lambda Exonuclease, a 5'-to-3' nuclease with a preference for 5' phosphorylated DNA (*Little, 1967*; *Mitsis and Kwagh, 1999*), the 5'- phosphorylated strand is degraded, leaving the non-phosphorylated strand as a ssDNA homology donor. This PCR-based approach has several advantages. First, the homology arms are included in the ~125 nt primers as 5' overhangs such that a single PCR generates the complete donor without any cloning steps. Second, the same primers can be used to amplify many different SIC templates for the same GOI. Third, the priming sequences and overall construct length do not change between genes. Hence, the same protocol can be applied to create homology donors for different genes. Fourth, the PCR-based method does not require bacterial transformation, eliminating possible rearrangements associated with propagation of DNA in bacteria (*Figure 1B*). Finally, the risk of inserting PCR-induced mutations in the SIC is negligible when a proofreading polymerase is used.

We first tested the efficacy of ssDNA drop-in constructs as a substrate for homology-directed repair in *Drosophila* S2R+ cells stably transfected with Cas9 (S2R+-MT::Cas9; *Viswanatha et al., 2018*). Specifically, we were interested in generating a collection of *Drosophila* S2R+ cells in which different organelles are marked by a protein tagged with a superfolder GFP (sfGFP) in each cell line (*Pédelacq et al., 2006*). We therefore amplified a SIC consisting of *attP-SA-Linker-sfGFP-Linker-SD-attP* (1392 nt including homology arms) using 20 different gene-specific primer pairs (*Figure 2A*, *Supplementary file 3*). Each gene-specific ssDNA homology donor was electroporated into Cas9-positive cells along with a corresponding gene-specific gRNA and then subjected to fluorescence-activated cell sorting (FACS) (*Figure 2B*).

The frequency of GFP+ cells was determined by FACS (*Figure 2—figure supplement 1*) to range between ~1–4%. GFP-positive cells were clonally isolated, cultured, and analyzed. We observed that

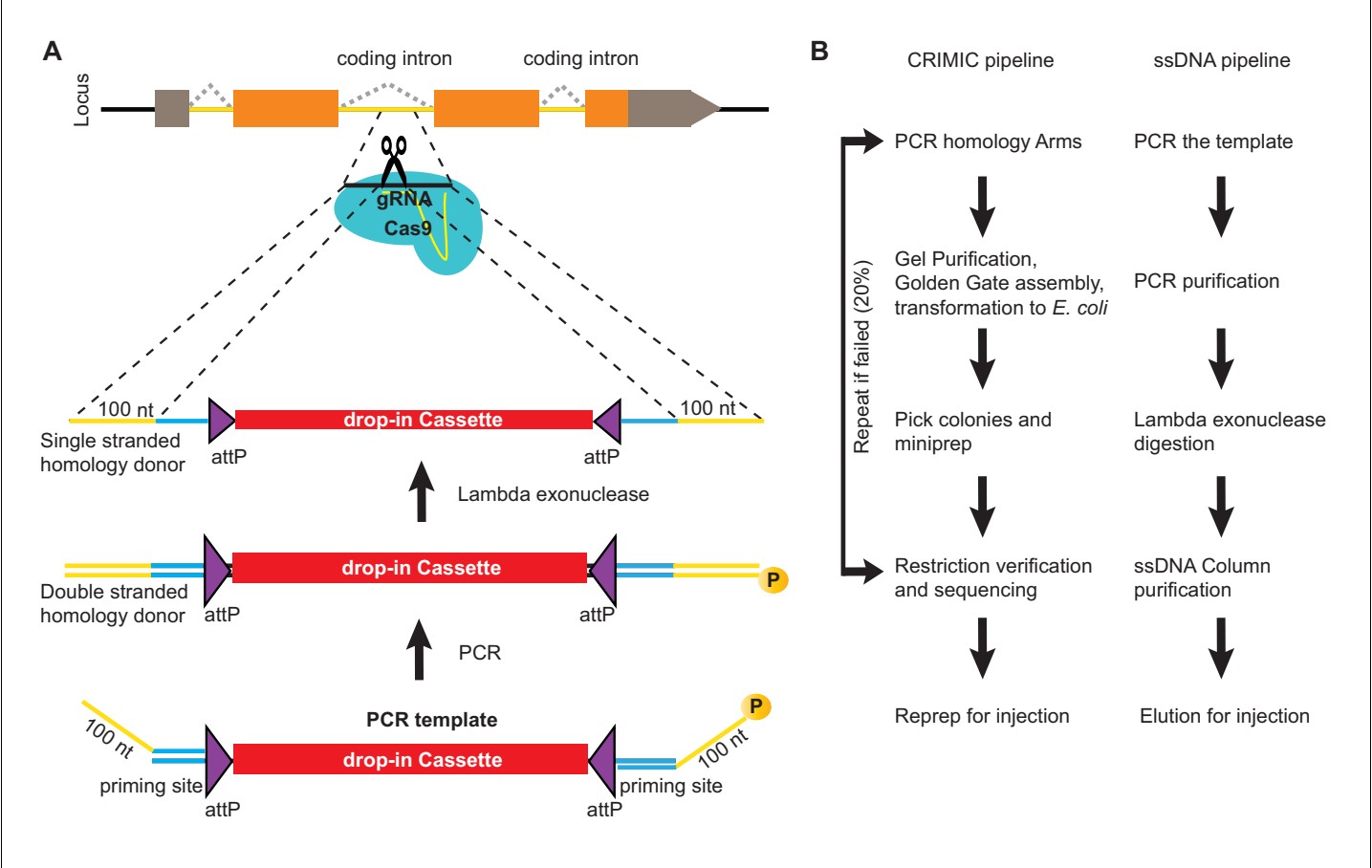

**Figure 1.** ssDNA pipeline is faster than cloning CRIMIC constructs. (**A**) Schematic of PCR-based generation of drop-in ssDNA constructs. Gray boxes, UTRs; orange boxes, coding exons; yellow line, coding introns; black line, outside coding introns and exons. (**B**) Comparison of donor generation pipelines for CRIMIC and PCR-based drop-in ssDNA homology donors. Making ssDNA donors is about 5X faster than making CRIMIC donors.
DOI: https://doi.org/10.7554/eLife.51539.002

for any given clone, all cells had the same subcellular GFP localization, indicating that they were derived from a single cell and that the insertion was stably integrated. For 19 out of 20 genes targeted we observed GFP signal by FACS (*Table 1*). For 12 of 19 genes, we could establish GFP+ clones, verify correct integration by PCR, and determine subcellular localization by immunostaining (*Table 1*, *Figure 2C*, *Figure 2—figure supplement 2*). With the exception of Ref2P, all correct insertions of GFP resulted in fusion proteins with the expected subcellular distribution. The genes for which we observed GFP signal by FACS but could not successfully isolate GFP+ clones tended to be expressed at low levels in S2R+ cells based on modEncode expression profiling (*Cherbas et al., 2011*; *Table 1*). For these genes it is possible that the signal-to-noise ratio for GFP was insufficient to robustly select GFP+ clones, leading to the loss of positive cells in the population.

Sequencing of the SIC insertion sites (*Table 1*) for the 12 cell lines showed that the insertions are precise. Western blotting of cell lysates confirmed that the inserted tags lead to GFP fusion proteins of the expected molecular weights (*Figure 2—figure supplement 3*). Given the dynamic localization of Polo protein during mitosis (*Llamazares et al., 1991*) we recorded the Polo subcellular localization pattern in live cells through mitosis. Time-lapse confocal imaging of Polo-GFP showed that the protein is localized to centrosomes, spindle, and midbody during cell division, in agreement with the data obtained by immunofluorescence (*Llamazares et al., 1991*) or using *polo-GFP* transgenes (*Moutinho-Santos et al., 1999*; (*Video 1*).

Previously, PCR generated double-stranded constructs containing GFP and an antibiotic resistance gene have been used for homologous recombination in S2 cells (*Böttcher et al., 2014*;

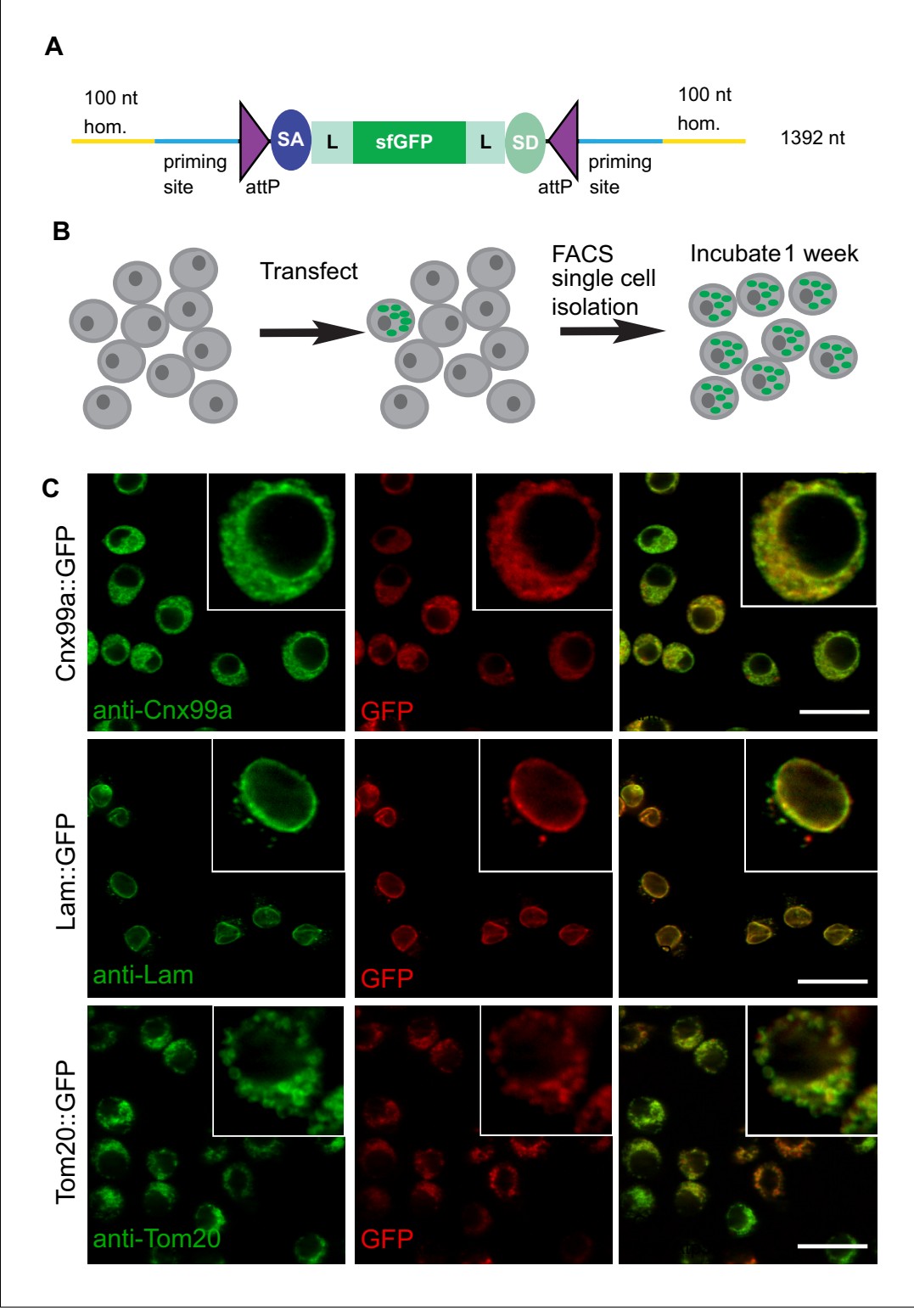

**Figure 2.** ssDNA homology donors are effective in S2 cells to tag organelles. (**A**) Schematic of drop-in cassette encoding for sfGFP artificial exon. Size of the construct including the homology arms is indicated on the right. sfGFP: superfolderGFP; SA: Splice Acceptor of *mhc*; SD: Splice Donor of *mhc*; L: flexible linker that consists of four copies of Gly-Gly-Ser. (**B**) Diagram of steps to isolate cell clones resulting from successful homologous recombination events. (**C**) Examples of S2R+ cells with organelles marked with GFP. Left panel, antibody staining; middle panel, GFP signal; right panel, the merge.

*Figure 2 continued on next page*

*Figure 2 continued*

DOI: https://doi.org/10.7554/eLife.51539.003

The following figure supplements are available for figure 2:

**Figure supplement 1.** FACS data of control cells (left) and ssDNA knock-in cells (right).

DOI: https://doi.org/10.7554/eLife.51539.004

**Figure supplement 2.** Detection of subcellular localization of GFP tagged proteins in S2 cells.

DOI: https://doi.org/10.7554/eLife.51539.005

**Figure supplement 3.** Western blot analysis of tagged proteins observed in S2R+ cells.

DOI: https://doi.org/10.7554/eLife.51539.006

*Kunzelmann et al., 2016*). However, selection with a drug resistance gene was used to enrich the population and GFP integration frequency was about 2% as judged by FACS. We were able to generate GFP protein traps in S2 cells using long ssDNA in up to 4% transfected cells and obtain clones without using drug-based selection (*Figure 2*, *Figure 2—figure supplement 2*).

**Table 1.** Summary of ssDNA drop-in mediated homologous recombination in S2R+ cells

| Organelle | Fly protein | clones obtained | population imaged | clones imaged | insertion sequence verified | Immunostained | Correct GFP localization | DGRC stock# | S2R+ expression (modENCODE RPKM) |
|---|---|---|---|---|---|---|---|---|---|
| Autophagosomes | Atg8a | 14 | ✓ | ✓ | N | N | N | N | 153 |
| Autophagosomes/ aggregates | Ref(2)P | 9 | ✓ | ✓ | Y | a-Ref2P, a-FK2 | N | N | 138 |
| Endoplasmic reticulum (ER) | Calnexin99A | 16 | ✓ | ✓ | Y | a-Cnx99a | Y | 273 | 235 |
| Endoplasmic reticulum (ER), transitional | Sec23 | 30 | ✓ | ✓ | Y | N | * | 294 | 101 |
| Endosomes, early | Rab5 | 9 | ✓ | ✓ | N | N | N | N | 98 |
| Endosomes, recycling | Rab11 | 23 | ✓ | ✓ | Y | N | * | 274 | 302 |
| G-Bodies (cytoplasmic puncta) | Pfk | 14 | ✓ | ✓ | N | N | N | N | 23 |
| Golgi (cis-Golgi) | Gmap | 10 | ✓ | ✓ | Y | a-GMAP | Y | 276, 277 | 15 |
| Golgi (trans-Golgi) | Sec71 | 10 | ✓ | ✓ | N | N | N | N | 8 |
| Golgi (trans-Golgi) | Golgin245 | 1 | ✓ | ✓ | Y | a-Golgin245 | Y | 280 | 27 |
| Kinetochore | Polo | 2 | ✓ | ✓ | Y | N | Y | 275 | 50 |
| Lipid droplets | Seipin | 12 | ✓ | ✓ | N | N | N | N | 9 |
| Lysosomes | spin | 2 | ✓ | ✓ | Y | a-Arl8 | Y | 293 | 112 |
| Lysosomes | Arl8 | 9 | ✓ | ✓ | Y | a-Arl8 | Y | 291 | 78 |
| Mitochondria | Tim17b | 3 | ✓ | ✓ | N | a-ATP5A | N | N | 266 |
| Mitochondria | Tom20 | 17 | ✓ | ✓ | Y | a-ATP5A | Y | 302 | 117 |
| Nuclear membrane, inner | dLBR | 0 | ✓ | N | N | N | N | N | 42 |
| Nuclear membrane, inner | Lamin | 53 | ✓ | ✓ | Y | a-Lamin | Y | 292 | 249 |
| Nucleolus | Fibrillarin | 14 | ✓ | ✓ | Y | a-Fib | Y | 278, 279 | 53 |
| Peroxisomes | Pmp70 | 6 | ✓ | ✓ | N | N | N | N | 26 |

\* = distribution was as expected, but no antibody available to test by co-stain

DOI: https://doi.org/10.7554/eLife.51539.007

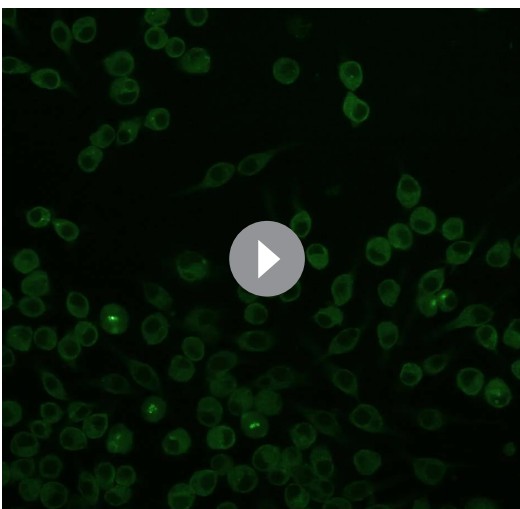

**Video 1.** Polo-sfGFP knock-in in S2R+ cells through ssDNA drop-in faithfully reports dynamic localization of Polo throughout the cell cycle.
DOI: https://doi.org/10.7554/eLife.51539.008

Previous studies by *Richardson et al. (2016)* analyzed the binding dynamics of Cas9 to target sites and showed an increase in homologous recombination efficiency by using homology donors with asymmetric homology arms. Whether the use of asymmetric homology arms will increase the knock-in efficiency remains to be tested for drop-in.

In summary, ssDNA drop-in constructs are simple to produce and provided an efficient homology substrate in S2R+ cells for about 60% of the tested genes. The generated cell lines with marked organelles—endoplasmic reticulum (2), recycling endosomes, cis and trans Golgi, kinetochores, lysosomes (2), mitochondria, nucleoli and nuclear envelopes (*Table 1*)—are listed at the *Drosophila* RNAi Screening Center (DRSC) website (https://fgr.hms.harvard.edu/crispr-modified-cell-lines) and available through the Drosophila Genomics Resource Center (DGRC; https://dgrc.bio.indiana.edu). These cell lines will be a useful resource for small- and large-scale studies of organelle biogenesis, organelle function, and/or subcellular distribution of organelles or specific fusion proteins, and genetic and pharmacological screens to identify regulators of organelle homeostasis.

## Integration of ssDNA drop-in donors in the germline

We next generated a drop-in SIC for in vivo Drosophila transformation that can be amplified using PCR primers. Our initial experiments showed that the empirical size limit for ssDNA production was ~2 kb, significantly smaller than the ~5 kb SIC present in the CRIMIC cassettes currently used by the GDP (*Lee et al., 2018a*). A minimal SIC should contain *attP* sites to enable downstream RMCE applications and a dominant marker that allows detection of positive insertions in the targeted locus. The smallest self-sufficient visible dominant marker in *Drosophila* is *3XP3-EGFP* (*Horn et al., 2000*). Hence, in order to remain under the size limit, we generated an *attP-3XP3-EGFP-attP* PCR template that is 1242 nt long and can be used for ssDNA donor (1442 nt including two 100 nt homology arms) generation (*Figure 3A*). The *attP-3XP3-EGFP-attP* cassette is not mutagenic unless inserted in exons and only generates a landing site for RMCE in the targeted region.

To test the efficiency of this new ssDNA drop-in construct in vivo, we compared the efficiency of the ssDNA drop-in donor (with 100 nt homology arms) and the current CRIMIC dsDNA donor (*attP-FRT-SA-T2A-GAL4-polyA-3XP3-EGFP-FRT-attP* with ~1 kb homology arms) for the same 10 genes using the same gRNAs by injecting ~400–600 embryos for each gene with each construct. Surviving adults were crossed with *y w* flies as single fly crosses and the numbers of independent targeting events were quantified. We targeted a set of 10 genes with each construct to have a large enough set to reduce the possibility of locus-specific properties skewing the results. We were successful in targeting four genes with both approaches and obtained insertions in *CG5009, CG9527, Khc,* and *NLaz* with the CRIMIC approach and *CG5009, endoB, Cp1,* and *Lst* with the ssDNA drop-in construct. We found that the use of CRIMIC constructs was more efficient, as judged by the number of independent fly lines produced per successful gene (*Figure 3B*).

Short ssDNA homology donors were previously used in *Drosophila* to integrate small epitope tags or site-specific recombination sites (*Gratz et al., 2013*; *Wissel et al., 2016*). Due to size constraints, these constructs lacked a dominant marker and detection of successful events relied on labor-intensive PCR strategies. More recently, novel methods were developed to generate and use longer ssDNAs (~1000 nt) as homology donors in vivo in mice (*Miura et al., 2015*; *Quadros et al., 2017*; *Lanza et al., 2018*). Our results show that longer ssDNA constructs with visible dominant markers integrate in the fly genome in vivo albeit with lower efficacy than double stranded homology donors with large homology arms (i.e. CRIMIC donors).

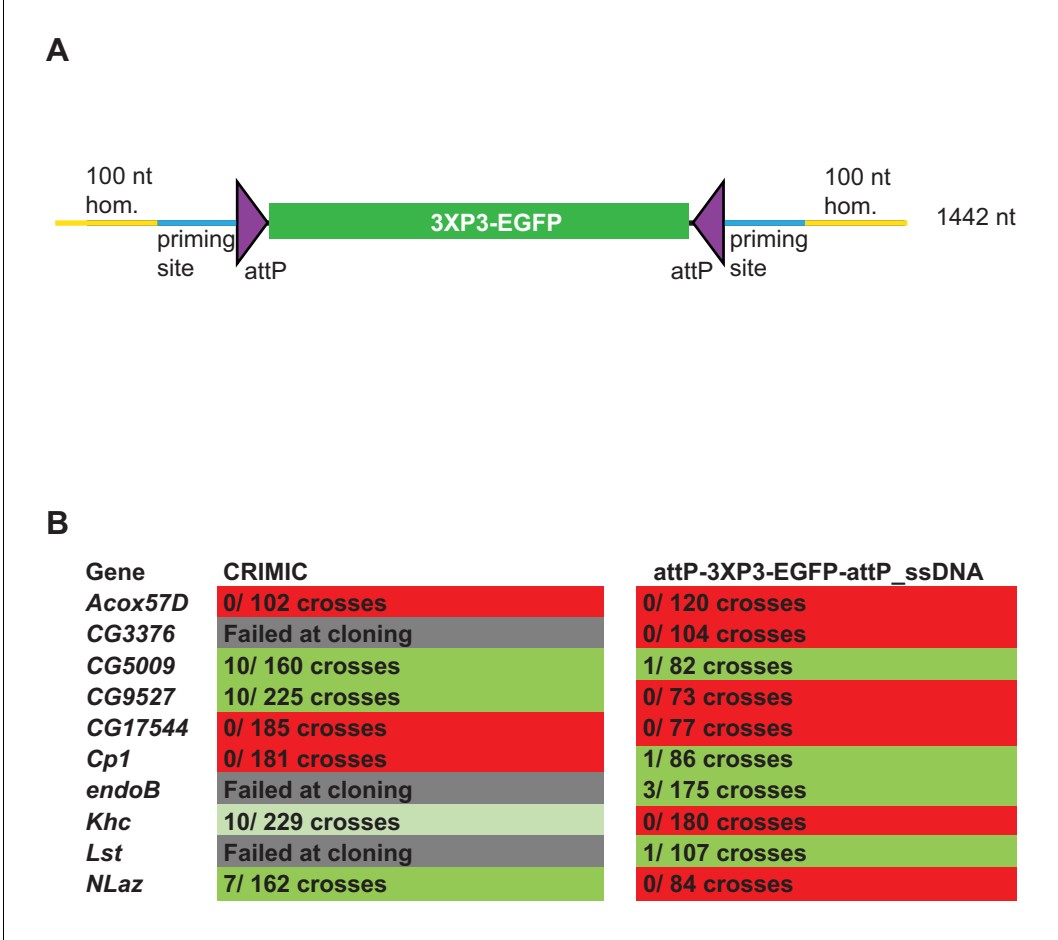

**Figure 3.** ssDNA constructs are not as efficient as double stranded CRIMIC constructs for fly transformation. (**A**) Schematic of drop-in cassette used for fly transformation. Size of the construct including homology arms is indicated on the right. (**B**) Injection results for the 10 genes selected for comparison of transformation efficacy. Numbers indicate positive events/fertile G0 single fly crosses. Red, no positive events; light green, positive non-confirmed events; dark green, genes with PCR-confirmed events.

DOI: https://doi.org/10.7554/eLife.51539.009

In summary, the ssDNA constructs are efficient donors for S2R+ cells for protein tagging by knock-in but for fly transgenesis they are not more efficient than the standard CRIMIC method. Nevertheless, the ease and low cost of producing these constructs and the transformation efficiency in S2 cells may justify their use to integrate GFP tags or landing sites in cultured cells or the germline.

## dsDNA drop-in donors of <2 kb are efficient homology donors for transgenesis

Given that the ssDNA drop-in constructs did not increase the success rate of fly transformation, we next attempted to optimize dsDNA homology donors for production ease and transformation efficiency. One means of increasing the targeting rate is to shorten the SIC, since homologous recombination is dependent on the size of the inserted cassette (*Beumer et al., 2013*). We used three strategies to shorten SICs (*Figure 4A*). First, we used a shortened GAL4 construct, referred to as miniGAL4 (1200 nt), which is about half the size of full-length GAL4 (2646 nt) and has about 50% of the transcriptional activity of full-length GAL4 in yeast (*Ding and Johnston, 1997*). Second, we used a very short dominant marker. Because *3XP3-EGFP* is ~1 kb in length, we opted to use an alternative marker that is only ~200 nt long and contains a ubiquitous U6 promoter expressing a gRNA sequence (gRNA1) that does not have a target in the fly genome. This strategy is based on the

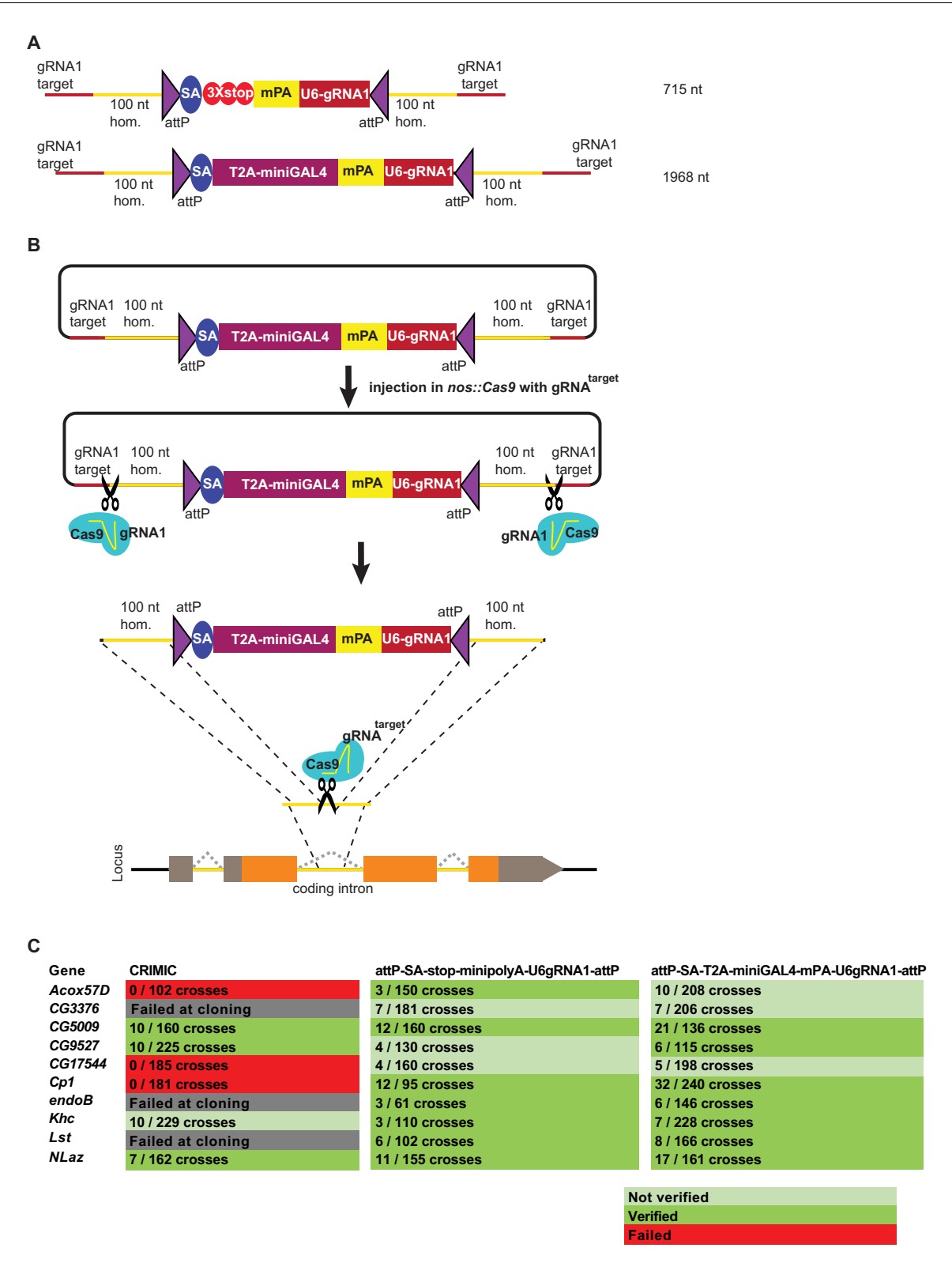

**Figure 4.** Double stranded DNA synthetic constructs are efficient for fly transformation. (**A**) Schematic of synthesized plasmid drop-in donors. mPA indicates *minipolyA*. (**B**) Schematic of the donor plasmid followed by linearization by Cas9 in germ cells and integration of donors in vivo. gRNA[target] is gene specific gRNA. (**C**) Injection results for the 10 genes selected for comparison of transformation efficacy. Numbers indicate positive events/fertile G0 single fly crosses. Red, no positive events; light green, positive non-confirmed events; and dark green, genes with PCR-confirmed events.
*Figure 4 continued on next page*

*Figure 4 continued*

DOI: https://doi.org/10.7554/eLife.51539.010

The following figure supplement is available for figure 4:

**Figure supplement 1.** Schematic and crossing scheme for CRISPR gRNA-based dominant marker strategy.

DOI: https://doi.org/10.7554/eLife.51539.011

single-strand annealing (SSA) pathway of DNA repair to reconstitute expression of a transgenic ubiquitous mCherry that is interrupted by the gRNA1 target sequence (*Figure 4—figure supplement 1*; *Garcia-Marques et al., 2019*). Upon Cas9-mediated double strand break, the modified non-functional *mCherry* gene is repaired and becomes functional, providing a visible marker. This SSA dependent repair reaction occurs in the F1 generation upon stable integration of U6:gRNA1 in the genome (*Figure 4—figure supplement 1*). Third, we shortened the polyA sequence from 135 nt to 35 nt (*minipolyA*) (*McFarland et al., 2006*). This resulted in *attP-SA-T2A-miniGAL4-minipolyA-U6gRNA1-attP*, which functions as a gene trap and is 1968 nt in length, including homology arms. We also generated a smaller minimal mutagenic construct, *attP-SA-3XSTOP-minipolyA-U6gRNA1-attP,* that is 715 nt in length including homology arms (*Figure 4A*). Both dsDNA drop-in constructs are small enough to be commercially synthesized at a low cost (less than $250).

To further improve the insertion efficiency, we decided to induce in vivo linearization of the plasmid constructs. Linearization has previously been shown to boost homologous recombination rates in cell culture, zebrafish and mouse transgenesis (*Cristea et al., 2013*; *Hisano et al., 2015*; *Suzuki et al., 2016*; *Yao et al., 2017*). Hence, use of short 100 nt left and right homology arms flanked by the gRNA1 target sites to linearize the construct in vivo upon injection may significantly increase the frequency of homologous recombination (*Figure 4B*).

It is worth noting that higher transgenesis rates have been reported in mice and worms using donor constructs linearized in vitro (*Paix et al., 2014*; *Paix et al., 2015*; *Paix et al., 2017*; *Dokshin et al., 2018*; *Yao et al., 2018*). However, in vitro linearized DNA or direct PCR products are very poor substrates for homologous recombination for germ line transformation in *Drosophila* (*Beumer et al., 2008*; *Böttcher et al., 2014*). We therefore opted to use dsDNA donors that are linearized in vivo by the Cas9 expressed in germ cells.

Upon injection of gene-specific gRNA coding plasmids and donor plasmids we isolated multiple transgenic lines for each of the 10 targeted genes. We obtained 65 independent transgenic lines for the *attP-SA-3XSTOP-minipolyA-U6gRNA1-attP* construct, and 119 independent transgenic lines for the *attP-SA-T2A-miniGAL4-minipolyA-U6gRNA1-attP* construct. We verified the insertion sites by genomic PCR in 7/10 genes for both methods. Analyses of the genomic DNA sequences of the three genes for which we could not verify the integration site revealed that these genes contain an unanticipated variation in the gRNA target site in the isogenized Cas9 injection stocks when compared to the FlyBase reference genome sequence (*dos Santos et al., 2015*; *Goodman et al., 2018*). This suggests that the precise homologous recombination rate is dependent on efficient cutting of the target site and is likely higher than 70%. Our current gRNA design platform considers these strain variations which can be substantial in introns.

Because we shortened the *polyA* tail from 135 nt (*Okada et al., 1999*) to 35 nt (*McFarland et al., 2006*), we assessed the mutagenic potential of new constructs that contain *minipolyA*. We used Western blotting with an antibody that recognizes the gene product of *CG5009* (*Drosophila* ortholog of *ACOX1*) to compare protein levels in animals homozygous for the CRIMIC allele (with 135 nt *polyA* tail) with protein levels of animals homozygous for either of the two constructs that carry the *minipolyA*. As shown in *Figure 5A*, the artificial exon *attP-SA-3XSTOP-minipolyA-U6gRNA1-attP* leads to ~40% decrease in protein levels and creates a much less severe allele than the CRIMIC allele. However, the *attP-SA-T2A-miniGAL4-minipolyA-U6gRNA1-attP* cassette leads to ~80–90% decrease in protein levels, similar to a CRIMIC insertion in the same locus (*Figure 5A*). The lower mutagenic efficacy of *attP-SA-3XSTOP-minipolyA-U6gRNA1-attP* may be the result of read through of stop codons, inefficient transcriptional stop at the *minipolyA* sequence or smaller size of the inserted artificial exon. The observation that *attP-SA-T2A-miniGAL4-minipolyA-U6gRNA1-attP* is potently mutagenic indicates that the combination of *SA-T2A* with *minipolyA* is stronger than *SA-3XSTOP-minipolyA*. Alternatively, the increased size of artificial exon in *attP-SA-T2A-miniGAL4-*

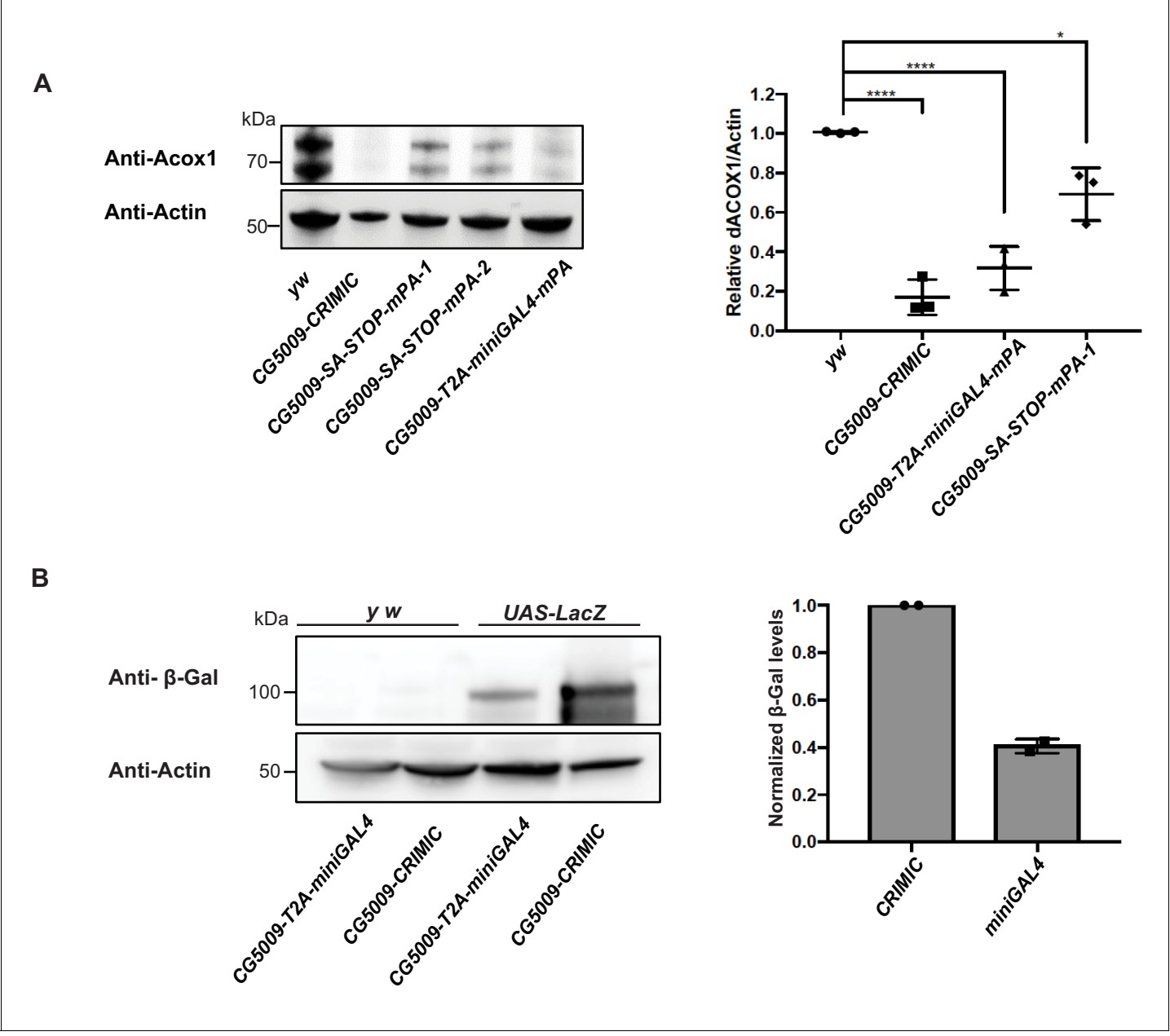

**Figure 5.** T2A-miniGAL4 gene trap cassette is mutagenic and expresses an active GAL4. (**A**) Western blot and quantification of the level of ACOX1 protein in flies homozygous for *SA-T2A-miniGAL4-minipolyA, SA-3XSTOP-minipolyA* or CRIMIC construct for *CG5009*. ****p<0.0001, *p<0.01. mPA1 and mPA2 indicate two independent lines of *CG5009-SA-3XSTOP-minipolyA*. (**B**) Western blot of β-galactosidase and quantification of heterozygous flies carrying a copy of the miniGAL4 construct compared to heterozygous flies carrying a CRIMIC.
DOI: https://doi.org/10.7554/eLife.51539.012

*minipolyA-U6gRNA1-attP* overcomes the limitation of *attP-SA-3XSTOP-minipolyA-U6gRNA1-attP* construct in mutagenesis efficacy.

The miniGAL4 construct had never been tested in flies. We therefore compared full length versus miniGAL4 induction of a *UAS-LacZ* reporter by Western blot. As shown in **Figure 5B**, we observed a ~ 60% reduction of reporter levels with T2A-miniGAL4 as compared to CRIMIC T2A-GAL4 driven *UAS-LacZ* expression for the *CG5009* locus (**Figure 5B**). Hence, the *attP-SA-T2A-miniGAL4-minipolyA-U6gRNA1-attP* drives lower levels of reporter expression when compared to CRIMIC, which may limit the use of miniGAL4 for genes expressed at low levels.

In summary, the decrease of SIC size and linearization increased the transgenesis rate compared to CRIMIC templates. We used novel selection markers, smaller *polyA* tails, and new GAL4 variants to decrease the size of the integrated construct. However, this decrease in size came with trade-offs. The U6-gRNA1 dominant marker is very easy to detect and is smaller than previously established dominant markers. However, this marker requires the presence of other transgenes for detection (*Figure 4—figure supplement 1*) and the reconstituted dominant marker transgene segregates independently from the targeted gene. These limitations make establishing and maintaining stocks more challenging. Moreover, T2A-miniGAL4 may not be strong enough to drive robust reporter expression in genes with low expression levels.

## Large double stranded drop-in dsDNA donors (∼5 kb) are efficient homology donors to integrate CRIMIC-like SICs

To avoid the issues raised in the previous section, we designed a strategy to integrate full length CRIMIC cassettes with short homology arms. To achieve this, we tested whether large DNA fragments with short homology arms (100 nt) could be integrated in target genes upon linearization using the gRNA1 in vivo. This would allow the use of dominant markers like *3XP3-EGFP* as well as integration of the full length *GAL4* gene with an extended *polyA* tail, that is the CRIMIC cassette SIC. Unfortunately, synthesis of a full length 5 kb CRIMIC cassette would be cost prohibitive (> $1,000) especially for thousands of genes. Hence, we developed a modified cloning strategy in which we first synthesize a *gRNA1 target-100nt homology arm-Restriction cassette-100nt homology arm-gRNA1 target* inserted into a pUC57 vector for each target gene (cost is $80). The SIC containing *attP-FRT-T2A-GAL4-polyA-3XP3-EGFP-FRT-attP* is then subcloned directionally into this plasmid in a single straightforward cloning step, replacing the restriction cassette with the SIC of interest (*Figure 6A*). We refer to these constructs *drop-in int100-CRIMIC* constructs. The SIC can be replaced by any other effector (e.g. *SA-GFP-SD*) to generate other homology donor constructs.

We injected vectors containing the full length CRIMIC cassette for seven of the genes in which we previously successfully inserted dsDNA drop-in cassettes. Using the *drop-in int100-CRIMIC* cassette we obtained multiple knock-in alleles in five genes, as verified by PCR (*Figure 6B*). Hence, 100 nt homology arms are sufficient to integrate large SICs into target sites. In addition, increasing homology arm length to 200 nt should not increase synthesis costs, as the total length of the construct remains less than 500 nt. Whether use of 200 nt may improve efficiency remains to be tested.

To ensure functionality of knock-in alleles, we compared expression patterns of drop-in int100 CRIMIC with the corresponding CRIMIC for *CG5009* (*Figure 6C*) as well as drop-in int100 CRIMIC with the *T2A-miniGAL4* for *Khc* (*Figure 6—figure supplement 1*) by crossing the flies to *UAS-NLS::mCherry* reporter lines. In both cases the constructs lead to mCherry expression in very similar patterns.

In vivo linearization was previously shown to lead to knock-ins in zebrafish and cell culture even when there is no homology arm, provided that the homology donor and target site are cut concomitantly (*Cristea et al., 2013*; *Auer et al., 2014*; *Schmid-Burgk et al., 2016*; *Suzuki et al., 2016*). This process is more prone to generating small deletions in the target site and the insertions are non-directional. Recently, a homology independent knock-in method was established for *Drosophila* cell culture and germline transformation (*Bosch et al., 2019*). By simultaneously cutting the donor construct and target region in the absence of homology arms, *Bosch et al. (2019)* integrated a CRIMIC cassette in 4 of 11 genes targeted. However, these insertion events were error-prone and non-directional. Note that for drop-in constructs, the presence of short homology arms allows the donor vectors to be synthesized cheaply and introducing larger SICs is accomplished by a single straightforward cloning step, providing a good balance between ease of construct generation and efficient in vivo use.

In summary, we have developed efficient pipelines for CRISPR knock-in using ssDNA in *Drosophila* cells or a dsDNA approach in the germline. With respect to the dsDNA donors, we significantly improved the overall efficiency compared to the method described in *Lee et al. (2018a)*. The previous method that required cloning large homology arm flanked cassettes has several pitfalls: homology arm PCRs often must be troubleshooted repeatedly; assembly products are often incorrect; and sequencing of the final product is often challenging and needs to be repeated to confirm the construct. Moreover, when the efficacy of fly transformation and the events that can be PCR verified on either side of knock-in region are included, the efficacy of CRIMIC constructs hover around 50% in

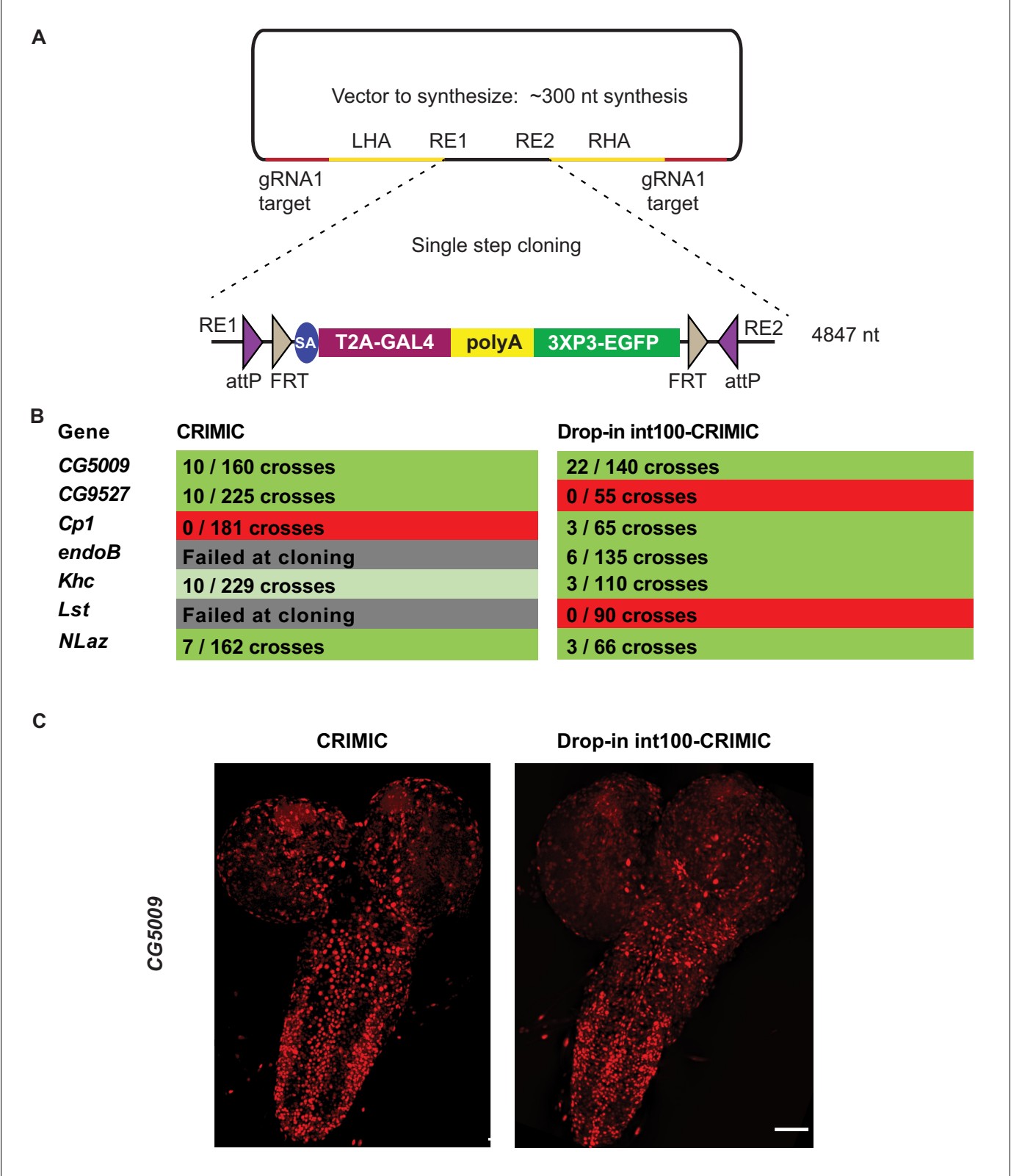

**Figure 6.** Single step cloning method allows efficient insertion of the CRIMIC cassette in coding introns. (**A**) Schematic of a single step cloning vector pUC57. LHA Left Homology Arm, RE1 Restriction Enzyme 1, RE2 restriction Enzyme 2, RHA Right Homology Arm. (**B**) Injection results for the seven genes selected to estimate transformation efficacy. Numbers indicate positive events/G0 single fly crosses. (**C**) Third instar larval brain expression
*Figure 6 continued on next page*

*Figure 6 continued*

domain of *CG5009* as determined by crossing conventional CRIMIC or drop-in int100-CRIMIC flies to *UAS-NLS-mCherry* reporter lines. Scale bar is 100 µm.

DOI: https://doi.org/10.7554/eLife.51539.013

The following figure supplement is available for figure 6:

**Figure supplement 1.** Comparison of expression domain obtained by T2A-miniGAL4 and drop-in int100-CRIMIC.

DOI: https://doi.org/10.7554/eLife.51539.014

optimal conditions. With the drop-in cassette strategy, not only is the cloning success rate nearly 100%, but the protocol requires little troubleshooting, dramatically reducing the bench time. Finally, factoring in the transformation and verification rate drop-in results in a 70–80% success rate. Key features that improved efficiency are: 1) shortening the homology arms to 100 nt, which allows synthesis followed by a simple cloning step, thus eliminating cloning failure and reducing the cost from ~$300 to ~$100 per construct; and 2) integration of target gRNA1 sites on either side of the SIC to linearize the donor with Cas9 in the germ cells. In conclusion, the methods we describe are efficient, simple, and precise.

## Materials and methods

### Generation of templates for ssDNA production

The sequences of scaffold vectors can be found in *Supplementary file 1*. Briefly, to generate the *attP-3XP3-EGFP-attP*, the scaffold vector 3XP3-EGFP cassette was amplified by PCR using the long primers 3XP3-EGFP-RI-for and 3XP3-EGFPrev-NotI and cloned between the *Eco*RI and *Not*I sites in pCasper3 (*Thummel and Pirrotta, 1992*). To generate *attP-SA-sfGFP-SD-attP*, a scaffold vector that we named pScaffold was produced by integrating annealed oligonucleotides with sequences M13For-attPfor-SbfI-AvrII-attPrev-M13rev in pCasper3 backbone in *Eco*RI-*Not*I sites. SA-sfGFP-SD was cloned as a three-fragment ligation in pScaffold with linker-SA (amplified from pDoubleHeader [*Li-Kroeger et al., 2018*]) with primers SA-for-Sbf and Linker-SA-rev_BamHI), sfGFP (amplified from pUAST-NLS-sfGFP-3XMyc-PEST with primers sfGFP-for_BamHI and sfGFP-rev_KpnI), and Linker-SD (amplified from pDoubleHeader with primers Linker-SD-for_KpnI and Linker-SD-rev_NotI).

### Production of ssDNA drop-in constructs

Gene-specific homology donors were produced by incorporating the homologous sites (regions spanning 100 nt upstream of 3 nt prior to the PAM (protospacer adjacent motif) for left homology arm or 97 nt downstream of that region for the right homology arm (*Supplementary file 1*) as 5' overhangs to primers with complementarity to the template (26 bases for the forward primer and 24 bases for the reverse primer). The long primers were ordered from IDT (Coralville, Iowa) as Ultramers. See *Supplementary file 3* for sequences of primers. The reverse primers were 5' phosphorylated. Four 50 µl PCR reactions were set up with Q5 Hot Start High-Fidelity 2X Master Mix (NEB #M0494L). PCR conditions were optimized using gradient PCR (Bio-Rad C1000 Touch). The optimal annealing temperature was 70°C. The elongation time was 1 min for *attP-3XP3-EGFP-attP* and *attP-SA-sfGFP-SD-attP*. Resulting PCR amplicons were pooled in two samples and isolated using Qiaquick spin columns (Qiagen #28106) following the manufacturer's protocol. Samples were eluted in 2 × 50 µl elution buffer from the kit. Two lambda exonuclease (NEB #M0262L) digestion reactions of 100 µl were set up using 4–6 µg DNA and 8 µl enzyme each. Digestion conditions were 37°C for 1 hr, followed by 10 min at 75°C for heat inactivation. Reaction products were pooled in two samples and ssDNA isolated using NEB Monarch DNA purification kit following the manufacturer's instructions (NEB #T1030L). Samples were isolated in 2 × 10 µl prewarmed (55°C) elution buffer from the kit and the DNA concentration was measured using NanoDrop One (ThermoFisher Scientific).

### Generation of GFP Knock-In cell lines

#### Cell culture and regular media

*Drosophila* cells stably expressing Cas9 (S2R+-MT::Cas9; Drosophila Genomics Resource Center cell stock #268; *Viswanatha et al., 2018*) were cultured in Schneider's Drosophila Medium 1X

(ThermoFisher Scientific #21720024) with 10% FBS and 1% penicillin/streptomycin (referred to as regular media). We note that this cell line is a derivative of S2R+ NPT005 (DGRC #229) and thus contains an mCherry tag in the *Clic* locus (*Neumüller et al., 2012*).

### Conditioned media
Conditioned media were prepared as previously described by *Housden et al. (2017)*.

### Cellular transfection
Electroporation of S2R+-MT::Cas9 cells was performed using a Lonza 4D Nucleofector (Lonza #AAF-1002B) following the manufacturer's protocol. For each transfection, 1 µg of sgRNA (100 ng) and 2 µL of the sfGFP donor (100 ng) were used with $4 \times 10^5$ sub-confluent S2R+-MT::Cas9 cells. See *Supplementary file 3* for all sgRNA, long primers and insert sequences. After electroporation, cells were immediately placed in regular media and health was monitored. Cell cultures were then maintained and expanded for FACS.

### Isolation of Single-Cell clones
Prior to FACS, $10^6$ cells were filtered through a 40 µM Falcon Cell Strainer (Corning #21008–949) into 15 mL conical tubes (ThermoFisher Scientific #14-959-70C). Single cells were then FACS-isolated on a BD Aria Ilu, based on the presence of GFP with an expression level greater than $2 \times 10^2$ (see *Figure 2—figure supplement 1*). Single cells were sorted into wells of a 96 well plate (VWR #29444–010) filled with 100 µL of conditioned medium. Cells were observed ~14 days later and any viable clones were expanded.

## Analysis of GFP Knock-In cell lines
### Image analysis
Multiple clonal cell lines were generated per gene and image analysis was used to measure the expression of the GFP knock-in marker. Several images of each live clone were taken using an InCell Analyzer 6000 automated confocal fluorescence microscope (GE Healthcare Lifesciences) using the dsRed channel to detect mCherry fluorescence present in all cells and the FITC channel to detect GFP fluorescence. Images were analyzed using CellProfiler (version 2.1.1). As mCherry is present throughout each cell, this image was used by CellProfiler to determine the outline of individual cells using the command 'Identify Primary Objects.' The outlines generated from this step were then applied to the GFP image through the process 'Identify Secondary Objects.' GFP fluorescence intensity was then measured using the command 'Measure Object Intensity,' which averages the GFP fluorescence within all of the cellular outlines (see *Figure 2—figure supplement 2*). Mean GFP intensities for each clone were noted, and the three clones per gene with the highest means were chosen for molecular analysis. All other clones were pooled and stored for use in a subsequent FACS if needed. Image data were managed using OMERO, as supported by the Harvard Medical School Image Data Management Core.

### Genomic DNA extraction
Genomic DNA was extracted from individual clones using a Quick-DNA MiniPrep Kit (Zymo Research #D3024) according to the manufacturer's protocol. Concentrations were measured using a Nano-Drop 8000 Spectrophotometer (ThermoFisher Scientific #ND-8000-GL).

### Molecular analysis
Due to the large size of the insertion, the 5' and 3' integration sites of each insertion were analyzed by PCR amplifying and sequencing a fragment spanning the junction of each end of the insertion with flanking DNA (see *Supplementary file 3* for a list the two primer sets per gene). These flanking sites were PCR amplified with High Fidelity Phusion Polymerase (NEB #M0530) using the following program: 1) 98℃ for 30 s, 2) 98℃ for 10 s, 3) 56℃ for 30 s, 4) 72℃ for 30 s (35 cycles), 5) 72℃ for 10 min, 6) 4℃ hold. PCR products were excised from a 2% TAE agarose gel and purified using a QIA-quick Gel Extraction Kit (Qiagen #28704) and Sanger sequenced at the Dana-Farber/Harvard Cancer Center DNA Resource Core.

## Immunostaining

Cells were fixed in 4% paraformaldehyde in phosphate-buffered saline with 0.1% Triton X-100 for 30 min. A standard staining protocol was used. Primary antibodies were used as indicated: Chicken anti-GFP (1:1000, Abcam ab13970), Mouse anti-Lamin (1:500, DSHB #ADL84.12), Mouse anti-Calnexin99A (1:5, DSHB #Cnx99A 6-2-1), Mouse anti-ATP5A (1:100, Abcam ab14748), Rabbit anti-Arl8 (1:500, DSHB Arl8), Goat anti-GMAP (1:2000, DSHB, #GMAP), Goat anti-Golgin245 (1:2000, DSHB, #Golgin245), Rabbit anti-Ref2P (1:500, Abcam #ab178440), Rabbit anti-Myc-tag (1:1000, Cell Signaling Technology #2278S). Secondary antibodies with Alexa Fluor conjugates and DAPI (Molecular Probes #D-1306) were used at 1:1000. Images were obtained using a GE IN Cell 6000 automated confocal microscope with a 60x objective. Time-lapse videos were generated by imaging every 30 s over a 2 hr period.

## Western blotting

Clones that are generated using phase one template contained a 5 bp deletion in the splice donor portion of the insert. To determine if full-length fusion proteins were generated, we analyzed the following: S2R+-MT::Cas9 (unmodified control), Actin::GFP (a GFP-positive control), *Rab11* GFP knock-in (no deletion in the insert control), *spin* GFP knock-in (5 bp deletion), two *Lam* GFP knock-in clones (5 bp deletion in the first, 206 bp deletion in the second), *Ref2p* GFP knock-in (5 bp deletion), and *Cnx99a* GFP knock-in (5 bp deletion). Cells were spun down and lysed by resuspension in Pierce RIPA Buffer (ThermoFisher Scientific #89901) and Halt Protease Inhibitor Cocktail, EDTA-Free (ThermoFisher Scientific #87785). Cells were agitated for 30 min at 4°C and then centrifuged at 15,000 rpm for 20 min at 4°C. The lysates (supernatant) were removed and held at −20°C. Protein concentrations were determined using a Pierce BCA Protein Assay Kit (ThermoFisher Scientific #23227) according to the manufacturer's protocol. Appropriate volumes of each lysate were added to 4X Laemmli Sample Buffer (BioRad #1610747), vortexed, held at 100°C for 10 min, and then spun down at 13,000 rpm for 3 min before loading into a Mini-PROTEAN TGX Precast Gel (BioRad #4561095) and running at 100 V for 1 hr. The gel was then transferred onto a PVDF membrane (BioRad #1620177) using the Trans-Blot Turbo Transfer System (BioRad #1704150). After blocking with 5% blocking solution and washing in TBST, the membrane was incubated in the primary antibody rabbit anti-GFP (1:5000; Molecular Probes #A6455) shaking at 4°C overnight. The membrane was then washed four times with TBST and incubated with Donkey anti-rabbit HRP (1:3000; GE Healthcare #NA934) for 1 hr at room temperature, washed with TBST, and prepared for imaging using the SuperSignal West Pico PLUS Chemiluminescent Substrate (ThermoFisher Scientific #34580) according to manufacturer's protocol. The blot was imaged using a ChemiDoc MP Imaging System (see *Supplementary file 3* top panel; BioRad #17001402). The blot was then stripped with Restore PLUS Western Blot Stripping Buffer (ThermoFisher Scientific #46430), reprobed with mouse anti-tubulin (1:2000; SigmaT #5168) and Sheep anti-mouse HRP (1:3000; GE Healthcare #NXA931), and re-imaged (see Supplementary Figure # middle panel). The blot was then stripped again and reprobed with mouse anti-lamin (1:500; DSHB #ADL84.12) and Sheep anti-mouse Horse Radish Peroxidase (HRP) (1:3000; GE Healthcare #NXA931), and re-imaged (see Supplementary Figure # bottom panel).

For western analysis in adult flies, flies were dissected and lysed in 0.1% CHAPS buffer [50 mM NaCl, 200 mM HEPES, 1 mM EDTA and protease inhibitor cocktail (Roche)]. Tissue or cell debris were removed by centrifugation. Isolated lysates were subjected to electrophoresis using a 4–12% gradient SDS-PAGE gel and transferred to Immobilon-FL polyvinylidene difluoride membranes. Loading input was adjusted for protein concentration. Primary antibodies used were as follows: Rabbit anti-ACOX1 (1:1000; HPA021195, Sigma), Rabbit anti-β-galactosidase (1:1000; MP Biomedicals #55976), and Mouse anti-Actin-c4 (1:5000; Millipore Sigma #MAB1501). Secondary antibodies include Jackson ImmunoResearch HRP conjugated (1:5000). Blots were imaged on a Bio-Rad ChemiDocMP. The intensity of each band was measured and normalized to a loading control using Imagelab software (Bio-RAD). Three biological replicates were performed and ordinary one way ANOVA was used to compare expression levels of ACOX1 in different conditions. Two technical replicates were performed for β-galactosidase measurements.

## Fly injections

ssDNA constructs were injected at 50–100 ng/µl concentration with 25 ng/µl gene specific gRNA encoding pCFD3 vector (*Port et al., 2014*). *attP-SA-3XStop-minipolyA-U6gRNA1-attP* and *attP-SA-T2A-miniGAL4-minipolyA-U6gRNA1-att* dsDNA drop-in constructs were injected at ~150 ng/µl concentration together with 25 ng/µl gene specific gRNA. dsDNA drop-in int100-CRIMIC constructs were injected at 300–400 ng/µl along with 25 ng/µl gene specific gRNA and 25 ng/µl pCFD3-gRNA1. Injections were performed as described in *Lee et al. (2018a)*. 400–600 *y w; iso; attP2(y+) {nos-Cas9(v+)}* embryos per genotype were injected. Resulting G0 males and females were crossed to *y w* flies as single fly crosses for *3XP3-EGFP* detection and with *actin5C-Cas9; actin5C-GF-gRNA2-FP; actin5C-mCherr-#1-ry* flies for gRNA1 detection (*Figure 4—figure supplement 1*; *Garcia-Marques et al., 2019*). Up to five independent lines were generated per construct per gene. *actin5C-GF-#2-FP* is an internal control that would detect non-specific activation of dominant markers. We have not detected GFP in any of the screened flies, showing the specificity of dominant marker detection through gRNA1.

## PCR validation

PCR primers that flank the integration site were designed for each targeted gene (*Supplementary file 3* for primer sequences). These primers were used in combination with insert-specific primers that bind 5' of the inserted cassette in reverse orientation and 3' of the insert in forward orientation (pointing outwards from the insert cassette). 200–800 nt amplicons were amplified from genomic DNA from individual insertion lines through single fly PCR (*Gloor et al., 1993*) using OneTaq PCR master mix (NEB #M0271L). PCR conditions were denaturation at 95℃ for 30 s, 95℃ 30 s, 58℃ 30 s, 68℃ 1 min for 34 cycles and 68℃ 5 min.

## dsDNA drop-in constructs production

Templates for ordering the dsDNA drop-in constructs can be found in *Supplementary file 2*. dsDNA drop-in constructs were ordered for production from Genewiz ('ValueGene' option) in pUC57 Kan vector backbone at 4 µg production scale. When lyophilized samples arrived from production, samples were resuspended in 25 µl of ddH$_2$O at 55℃ for 30 min. 19 µl was mixed with 1 µl gene-specific gRNA plasmid (25 ng/ul final concentration of gRNA plasmid). The rest was stored at −20℃ for back-up purposes.

## Confocal imaging of transgenic larval brains

Dissection and imaging were performed following the protocols in *Lee et al. (2018a)*. In brief, fluorescence-positive 3$^{rd}$ instar larvae were collected in 1x PBS solution and then cut in half and inverted to expose the brain. Brains were transferred into 1.5 mL eppendorf tubes and fixed in 4% PFA in 1xPBS buffer for 20 min. Brains were then washed for 10 min three times in 0.2% PBST. Finally, samples were mounted on glass slides with 8 µL of VectaShield (VectorLabs #H-1000) and imaged at 20x zoom with a Nikon W1 dual laser spinning-disc confocal microscope.

## Acknowledgements

The Drosophila Gene Disruption Project is supported by NIH NIGMS R01GM067858 to HJB. Confocal microscopy at the NRI is supported by the Neurovisualization Core of the IDDRC, funded by NIH U54HD083092. We thank the Harvard Medical School Image Data Management Core for support with the OMERO platform and the Harvard Medical School Division of Immunology's Flow Cytometry Facility for the use of their FACS equipment. We also thank Raghuvir Viswanatha, Justin Bosch, Denise Lanza and Jason Heaney for helpful discussions, Robert Levis for critical reading and editing of the manuscript and Tzumin Lee for fly stocks. Development of the tagged cell line resource at the Drosophila RNAi Screening Center (DRSC) was supported by NIH ORIP R24 OD019847 (PI: NP, Co-PI: A Simcox, Co-I: SEM). Additional relevant support for the DRSC includes NIH NIGMS R01 GM084947 and P41 GM132087 (PI: NP, Co-I: SEM). NP, ACS, and HB are investigators of Howard Hughes Medical Institute.

## Additional information

### Competing interests
Hugo J Bellen: Reviewing editor, *eLife*. The other authors declare that no competing interests exist.

### Funding

| Funder | Grant reference number | Author |
|---|---|---|
| NIGMS | R01GM067858 | Hugo J Bellen |
| Howard Hughes Medical Institute | | Norbert Perrimon<br>Allan C Spradling<br>Hugo J Bellen |

The funders had no role in study design, data collection and interpretation, or the decision to submit the work for publication.

### Author contributions
Oguz Kanca, Conceptualization, Data curation, Formal analysis, Supervision, Validation, Investigation, Methodology, Writing—original draft, Project administration, Writing—review and editing; Jonathan Zirin, Conceptualization, Data curation, Methodology, Project administration, Writing—review and editing; Jorge Garcia-Marques, Conceptualization, Resources, Writing—review and editing; Shannon Marie Knight, Ming Ge, Data curation, Validation, Methodology; Donghui Yang-Zhou, Zhongyuan Zuo, Liwen Ma, Data curation, Visualization, Methodology; Gabriel Amador, Hyunglok Chung, Ying Fang, Data curation, Methodology; Yuchun He, Wen-Wen Lin, Data curation, Supervision, Methodology; Shinya Yamamoto, Conceptualization, Writing—review and editing; Karen L Schulze, Resources, Project administration, Writing—review and editing; Yanhui Hu, Resources, Software, Methodology; Allan C Spradling, Funding acquisition, Project administration, Writing—review and editing; Stephanie E Mohr, Conceptualization, Data curation, Investigation, Methodology, Writing—review and editing; Norbert Perrimon, Supervision, Funding acquisition, Project administration, Writing—review and editing; Hugo J Bellen, Supervision, Funding acquisition, Writing—original draft, Project administration, Writing—review and editing

### Author ORCIDs
Oguz Kanca https://orcid.org/0000-0001-5438-0879
Karen L Schulze https://orcid.org/0000-0002-1368-729X
Allan C Spradling http://orcid.org/0000-0002-5251-1801
Stephanie E Mohr http://orcid.org/0000-0001-9639-7708
Norbert Perrimon http://orcid.org/0000-0001-7542-472X
Hugo J Bellen https://orcid.org/0000-0001-5992-5989

### Decision letter and Author response
Decision letter https://doi.org/10.7554/eLife.51539.023
Author response https://doi.org/10.7554/eLife.51539.024

## Additional files

### Supplementary files
• Supplementary file 1. Sequences of drop-in constructs.
DOI: https://doi.org/10.7554/eLife.51539.015

• Supplementary file 2. Protocol for designing drop-in int100-CRIMIC constructs.
DOI: https://doi.org/10.7554/eLife.51539.016

• Supplementary file 3. List of oligos and ultramers used in this study.
DOI: https://doi.org/10.7554/eLife.51539.017

• Transparent reporting form DOI: https://doi.org/10.7554/eLife.51539.018

## Data availability

All the fly lines and cell lines generated in this manuscript will be made available through Bloomington Drosophila Stock center and Drosophila Genomics Resource Center.

The following previously published dataset was used:

| Author(s) | Year | Dataset title | Dataset URL | Database and Identifier |
|---|---|---|---|---|
| Li-Kroeger D, Kanca O, Lee PT, Cowan S, Lee MT, Jaiswal M, Salazar JL, He Y, Zuo Z, Bellen HJ | 2018 | An expanded toolkit for gene tagging based on MiMIC and scarless CRISPR tagging in Drosophila - sequence files | https://zenodo.org/record/1341241 | Zenodo, 1341241 |

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
