## [Decision Letter]

**Acceptance summary:**

This is an excellent study and the revisions address the points raised by the reviewers satisfactorily. This work is part of a larger effort to create a gene disruption line for every gene in *Drosophila* that has a human homolog. The strategy has been to generate interruptions in the introns of genes. These interruptions consist of a 'swappable cassette' due to the presence of attp sites. Nested within the two attp sites are various payloads such as a GFP (with splice acceptor and donor), resulting a 'protein trap', or a T2A-Gal4 with a PolyA tail similarly that interrupts translation. You have generated about 1500 such MiMIC and CRIMIC lines out of the 10,000 that they intend to make. Given the scale of the endeavour, both cost and efficiency of transgenesis are important considerations. The current methods are expensive (1500 USD per line), and because they involve multiple cloning steps and injections of large constructs into embryos, the effective efficiency is low (50%).

**Decision letter after peer review:**

Thank you for submitting your article "An efficient CRISPR-based strategy to insert small and large fragments of DNA using short homology arms" for consideration by *eLife*. Your article has been reviewed by three peer reviewers, including K VijayRaghavan as the Reviewing Editor, Senior Editor, and Reviewer #1.

The reviewers have discussed the reviews with one another and the Reviewing Editor has drafted this decision to help you prepare a revised submission.

Summary:

This is an excellent study and should be published after revision. This work is part of a larger effort to create a gene disruption line for every gene in *Drosophila* that has a human homologue. Their strategy has been to generate interruptions in the introns of genes. These interruptions consist of a 'swappable cassette' due to the presence of attp sites. Nested within the two attp sites are various payloads such as a GFP (with splice acceptor and donor), resulting a 'protein trap', or a T2A-Gal4 with a PolyA tail similarly that interrupts translation. They have generated about 1500 such MiMIC and CRIMIC lines out of the 10,000 that they intend to make. Given the scale of their endeavour, both cost and efficiency of transgenesis are important considerations. The current methods they use are expensive (1500 USD per line), and because they involve multiple cloning steps and injections of large constructs into embryos, the effective efficiency is low (50%). They have now tried multiple things to reduce cost and retain/enhance the efficiency of transgenesis and in this manuscript, they lay all this tinkering out for us.

They first attempted a cloning free, PCR based strategy that utilised universal primers combined with a 100 nucleotide homology arm overhang to generate single-stranded DNA donors for their 'drop-in' cassettes. They tested this strategy with 20 genes in S2 cells, and 10 in germline transformations. They were able to generate stable S2 lines for 12 (of 20) of the genes and these they verified to be accurate in terms of their precision of insertion, and subcellular localisation of the resultant GFP fusion protein. Unfortunately, this simple and cheap strategy had much lower efficiency in germline transgenesis that the previous methods.

So they turned to tinker with a few aspects of the previous strategy, namely: reducing the size of the selectable marker; reducing the size of the polyA tail that interrupts translation; reducing the Gal4 to a 'miniGal4'; linearising the injected plasmid in vivo; shortening the size of the homology arm (100nt as opposed to 1000nt); alternate cloning strategy.

Although these were cheaper to synthesise and gave excellent efficiencies in transformation, the miniGal4 was far less active, the miniPolyA far less effective in blocking translation and the mini-selectable marker segregated away. Finally the authors determine that the old CRIMIC cassettes, but with shorter homology arms generated via an alternate cloning strategy and an in vivo linearisation of the plasmid provided the right balance between cost (1/3 the previous cost) and efficiency (though I can't tell what their efficiency is!)

This is a paper about tinkering with various strategies and it's precisely because of this that we would recommend it: It lets you look under the hood. So much work goes into such an endeavour, and we are pleased that they are going to put this out. It's not just nice, but will inform many other systems where transgenesis using similar strategies are being attempted. There's a wealth of rigorously documented what-works-what-doesn't knowledge here that typically people don't talk about.

Essential revisions:

1) For the more laborious, double-stranded DNA (dsDNA) synthetic constructs a significantly improved frequency of HDR is nicely demonstrated. However, the results for the single-stranded DNA homology donors (ssDNA) are as persuasive. The authors may want to consider repeating these experiments (in cell culture and in flies) using the approach published by Richardson and colleagues, 2016. These authors presented an asymmetric ssDNA donor design that increased the rate of HDR in human cells up to 60%. The design is based on their observation that after DNA cleavage, Cas9 first releases the PAM-distal nontarget strand before dissociation. This preferential release of the cleaved DNA seems to offer a privileged entry point for the ssDNA donor which is accounted for in an asymmetric design of the ssDNA donor (36 and 91 nucleotides homology arms, respectively). Exploring this technology in the context of *Drosophila* has the potential to enhance the method described by Kanca and colleagues (as already pointed out in the review by Bier et al., 2018, as well as more profoundly benchmarking their technology beyond Lee et al., 2018. We understand that this could take time and, unless the authors have already embarked on these directions, the authors should discuss the strengths and limitations of their results for ssDNA homology donors.

2) The shortened double-stranded Swappable Integration Cassette (SIC) design comes with the cost of losing the marker 3xP3-EGFP. The 3xP3-EGFP cassette is a direct measure of HDR efficiency. The replacement is the expression of a sgRNA used for the repair of a non-functional m-Cherry reporter. In this case, reporter activation is used as a co-CRISPR strategy and does not directly reflect HDR efficiency. Co-CRISPR strategies using recessive markers have been described by: Ge et al., 2016 and Kane et al., 2017.

First, the replacement of the 3xP3-EGFP marker should be described as a co-CRISPRing strategy in the text, citing the papers that developed and used this strategy previously. Second, it is unclear to the reader if the dominant marker (CRISPR-based mCherry repair) indeed has been used to establish fly lines described in the paper. Were the flies listed in Figure 4C pre-selected based on mCherry expression? How many of the non-mCherry lines were nevertheless successfully modified in the GOI? These points should be mentioned more clearly. In case it has been used, it would be interesting to know how many mCherry-positive animals resulted in successfully targeted GOIs. If data is available please add it to the supplementary information.

3) The authors make a point that their drop-in int100 CRIMIC approach to germline transformation significantly improves the methods previously used by the GDP. However, the improvement appears to be primarily due to the reduction of "cloning failures." What are "cloning failures?" Are they what most labs deal with on a routine basis by adjustments in the protocol? One can imagine that for a large-scale project avoiding such failures is highly desirable. For an individual lab, it may also be a convenience-but not necessarily a big deal. In the absence of any discussion of what a "cloning failure" is and how it might otherwise be addressed, it is hard to know whether the improvement obtained with the drop-in int100 CRIMICs is anything more than incremental.

4) The authors show that there is a reduced translation product from an interrupted gene when the miniGal4 construct is used vs. a simple STOP cassette. They discuss this difference in the context of the mPA, but the mPA seems unlikely to be the cause of the diminished expression since both constructs have it. Do the authors suppose that the sequence of the inserted construct (STOP vs. minGal) affects the mPA efficacy? A more likely alternative is that the two constructs differentially affect SA function. Do the authors have reason to rule out this possibility?

5) For the generation of ssDNA donors the authors describe using 100 nt extensions of the PCR primers. This is a very nice idea, but it would be useful to know how much this reduces the efficiency of PCR amplification and/or gives rise to non-specific products?

6) The manuscript refers to "gRNA1" but it seems unlikely to be a single sequence, otherwise the U6gRNA1 sequence would be a target of the guide RNA it produces. It looks from the information provided that the gRNA1 is actually two (slightly different) sequences designed to avoid cross-reaction when the "U6gRNA1" is expressed. This ambiguity is confusing and if the interpretation given here is correct it would be best to assign different names to the two sequences. Also, it would be useful if the authors showed a picture of an animal expressing the U6gRNA1-activated mCherry selection marker. It seems that this marker will be ubiquitously expressed and the authors should probably comment on the potential complication of using a marker that will interfere (until removal) with any other fluorescent label in the red channel.

7) Several figure and table legends are short on detail, in some cases making interpretation difficult. For example in Table 1, what does "N/A" mean? Particularly for "Insertion sequence-verified?" It seems like it either it was (Y) or wasn't (N). Also, in Figure 3B (and elsewhere) what does "Failed at Cloning" mean? And what does it mean that an event is "non-confirmed." In Figure 5A, what is the difference between mPA-1 and mPA-2? Were two versions of the mPA made?

---

## [Author Response]

Essential revisions:1) For the more laborious, double-stranded DNA (dsDNA) synthetic constructs a significantly improved frequency of HDR is nicely demonstrated. However, the results for the single-stranded DNA homology donors (ssDNA) are as persuasive. The authors may want to consider repeating these experiments (in cell culture and in flies) using the approach published by Richardson and colleagues, 2016. These authors presented an asymmetric ssDNA donor design that increased the rate of HDR in human cells up to 60%. The design is based on their observation that after DNA cleavage, Cas9 first releases the PAM-distal nontarget strand before dissociation. This preferential release of the cleaved DNA seems to offer a privileged entry point for the ssDNA donor which is accounted for in an asymmetric design of the ssDNA donor (36 and 91 nucleotides homology arms, respectively). Exploring this technology in the context of Drosophila has the potential to enhance the method described by Kanca and colleagues (as already pointed out in the review by Bier et al., 2018, as well as more profoundly benchmarking their technology beyond Lee et al., 2018. We understand that this could take time and, unless the authors have already embarked on these directions, the authors should discuss the strengths and limitations of their results for ssDNA homology donors.

We thank to reviewers for this suggestion. When Richardson and colleagues published their work we tried the asymmetric homology arm constructs for one of the targeted genes in S2 cells as well as in fly transformation. In both assays we did not notice a difference in efficiency for knock-in in Rab11 locus. However, since we did not perform a systematic study for multiple genes we did not discuss these preliminary results in our manuscript. We can mention this in the Discussion if the reviewer wishes, but we prefer not to do this as we can only speculate as to what the data mean. We therefore suggest that asymmetry may increase the efficiency of single stranded transformation, referring to Richardson et al. paper. We added the paragraph **“**Previous studies by Richardson et al. (Richardson et al., 2016) analyzed the binding dynamics of Cas9 to target sites and showed increase in homologous recombination efficiency by using homology donors with asymmetric homology arms. Whether the use of asymmetric homology arms will increase the knock-in efficiency remains to be tested for Drop-in.” Note that given that a cassette with 3XP3-GFP combined with the full length GAL4 is more desirable for fly transformation, these constructs would be much too large to incorporate in ssDNA donors with any of the currently established methods.

2) The shortened double-stranded Swappable Integration Cassette (SIC) design comes with the cost of losing the marker 3xP3-EGFP. The 3xP3-EGFP cassette is a direct measure of HDR efficiency. The replacement is the expression of a sgRNA used for the repair of a non-functional m-Cherry reporter. In this case, reporter activation is used as a co-CRISPR strategy and does not directly reflect HDR efficiency. Co-CRISPR strategies using recessive markers have been described by: Ge et al., 2016 and Kane et al., 2017.First, the replacement of the 3xP3-EGFP marker should be described as a co-CRISPRing strategy in the text, citing the papers that developed and used this strategy previously. Second, it is unclear to the reader if the dominant marker (CRISPR-based mCherry repair) indeed has been used to establish fly lines described in the paper. Were the flies listed in Figure 4C pre-selected based on mCherry expression? How many of the non-mCherry lines were nevertheless successfully modified in the GOI? These points should be mentioned more clearly. In case it has been used, it would be interesting to know how many mCherry-positive animals resulted in successfully targeted GOIs. If data is available please add it to the supplementary information.

We appreciate the suggestion from the reviewer. However, the gRNA1 used in our constructs as a dominant marker is not used in a Co-CRISPR context. The injected flies do not contain the Actin-mcherr-#1-ry marker that gets repaired by single strand annealing. This repair happens in the next generation in the presence of U6:gRNA1 that is integrated in the genome together with the SIC. When the U6:gRNA1 is not integrated in the genome of F1 flies, Actin-Cas9 does not result in the repair of Actin-mcherr-#1-ry. Hence, the U6:gRNA1 behaves like a dominant marker, and not like a co-CRISPR gRNA. We have clarified this in the text by adding “This SSA dependent repair reaction occurs in the F1 generation upon stable integration of U6:gRNA1 in the genome (Figure 4—figure supplement 1).”

3) The authors make a point that their drop-in int100 CRIMIC approach to germline transformation significantly improves the methods previously used by the GDP. However, the improvement appears to be primarily due to the reduction of "cloning failures." What are "cloning failures?" Are they what most labs deal with on a routine basis by adjustments in the protocol? One can imagine that for a large-scale project avoiding such failures is highly desirable. For an individual lab, it may also be a convenience-but not necessarily a big deal. In the absence of any discussion of what a "cloning failure" is and how it might otherwise be addressed, it is hard to know whether the improvement obtained with the drop-in int100 CRIMICs is anything more than incremental.

Based on years of troubleshooting the cloning protocol, for a given batch of 96 designs, we are able to amplify ~90% homology arms by PCR. We then ligate the homology arms, insertion cassette, and backbone by golden gate assembly. The best-case scenario for a given batch is ~90% successful assembly, for an overall cloning efficiency of 80%. We must emphasize that this is our highest possible success rate, and depending on the batch, we sometimes can only achieve ~50% cloning efficiency. There is also a tremendous amount of bench time that goes into successfully cloning large homology arm flanked cassettes and many areas where the cloning can fail: homology arm PCRs must be troubleshooted repeatedly; many of the assembly products are incorrect; and sequencing of the final product is often challenging and needs to be repeated several times to confirm the construct. For a lab that does not specialize in making such constructs, the failure rate and cost in time is likely even greater. With the drop-in cassette strategy, not only is the cloning success rate nearly 100%, but the protocol requires little troubleshooting, dramatically reducing the bench time.

We explain the sources of cloning failures by adding the section “The previous method required cloning large homology arm flanked cassettes has several pitfalls: homology arm PCRs often must be troubleshooted repeatedly; assembly products are often incorrect; and sequencing of the final product is often challenging and needs to be repeated to confirm the construct. […] With the drop-in cassette strategy, not only is the cloning success rate nearly 100%, but the protocol requires little troubleshooting, dramatically reducing the bench time. The transformation and verification rate result in a 70-80% success rate.”

4) The authors show that there is a reduced translation product from an interrupted gene when the miniGal4 construct is used vs. a simple STOP cassette. They discuss this difference in the context of the mPA, but the mPA seems unlikely to be the cause of the diminished expression since both constructs have it. Do the authors suppose that the sequence of the inserted construct (STOP vs. minGal) affects the mPA efficacy? A more likely alternative is that the two constructs differentially affect SA function. Do the authors have reason to rule out this possibility?

The difference in the efficiency of STOP vs. miniGAL4 is indeed puzzling. We believe that T2A in the miniGAL4 acts as a stronger translational abort signal than the stop codons. In the presence of a strong transcriptional stop signal such as the full-length SV40 polyA transcriptional stop compensates for the stop codon read through. Although the splice acceptor sequence is identical between SA-STOP and SA-T2AminiGAL4 we cannot rule out that they affect splicing at different levels. One possible explanation for possible decreased SA efficacy is the size of SIC that we integrate. We now add to the Discussion “The lower mutagenic efficacy of *attP-SA-3XSTOP-minipolyA-U6gRNA1-attP* may be the result of read through of stop codons, inefficient transcriptional stop at minipolyA sequence or smaller size of inserted artificial exon. […] Alternatively, the increased size of artificial exon overcomes the limitation of *attP-SA-3XSTOP-minipolyA-U6gRNA1-attP* construct in mutagenesis efficacy”.

5) For the generation of ssDNA donors the authors describe using 100 nt extensions of the PCR primers. This is a very nice idea, but it would be useful to know how much this reduces the efficiency of PCR amplification and/or gives rise to non-specific products?

After the optimization of annealing temperature through gradient PCR, our protocol does not suffer from noticeable decrease in PCR yield or from non-specific bands.

6) The manuscript refers to "gRNA1" but it seems unlikely to be a single sequence, otherwise the U6gRNA1 sequence would be a target of the guide RNA it produces. It looks from the information provided that the gRNA1 is actually two (slightly different) sequences designed to avoid cross-reaction when the "U6gRNA1" is expressed. This ambiguity is confusing and if the interpretation given here is correct it would be best to assign different names to the two sequences.

gRNA1 sequence is the same between the target and U6:gRNA1. The difference is the presence of PAM sequence in gRNA1 target sites and lack of it in U6:gRNA1 construct. Since CRISPR/Cas9 cut is dependent on the PAM sequence U6:gRNA1 is not cut by the gRNA is produces.

Also, it would be useful if the authors showed a picture of an animal expressing the U6gRNA1-activated mCherry selection marker. It seems that this marker will be ubiquitously expressed and the authors should probably comment on the potential complication of using a marker that will interfere (until removal) with any other fluorescent label in the red channel.

U6gRNA1-activated mCherry selection marker is indeed ubiquitously expressed but as indicated in the last paragraph of the subsection “dsDNA drop-in donors of ˂ 2kb are efficient homology donors for transgenesis”, this reconstituted actin-mCherry segregates independently from the insert. Selecting flies with knock-in allele on a balancer that do not inherit the actin-mCherry transgene on the different chromosome will ensure that it will not interfere with downstream applications. The Figure 4—figure supplement 1 contains the crossing scheme to ensure proper balancing while removing the actin-mCherry transgene from the background. Moreover, as discussed in the following section, use of int100-CRIMIC constructs remove these complications.

7) Several figure and table legends are short on detail, in some cases making interpretation difficult. For example in Table 1, what does "N/A" mean? Particularly for "Insertion sequence-verified?" It seems like it either it was (Y) or wasn't (N). Also, in Figure 3B (and elsewhere) what does "Failed at Cloning" mean? And what does it mean that an event is "non-confirmed." In Figure 5A, what is the difference between mPA-1 and mPA-2? Were two versions of the mPA made?

We appreciate the reviewers pointing this out to us. In Table 1, we used N/A to indicate where it was not relevant to evaluate the data (for example, if there were no clones, we could not sequence-verify them). However, this is unnecessarily complicated, so we have changed the table to a binary Y/N in each category, unless otherwise noted. In Figure 5A mPA1 and mPA2 are two independent fly lines that result from the injection of same mPA construct. We now indicate this in the Figure 5 legend.